# PERSPECTIVE

## When pain is not the same: baroreceptor activation delineates chronic pain and pain-free participants' perception and modulation of noxious pressure

Luke A. Henderson 

*School of Medical Sciences (Neuroscience), Brain and Mind Centre, University of Sydney, Australia*

Email: luke.henderson@sydney.edu.au

Handling Editors: Richard Carson & Vaughan Macefield

The peer review history is available in the Supporting Information section of this article (https://doi.org/10.1113/JP287685#support-information-section).

Investigations into the perception and modulation of pain often focus on ascending and descending pain pathways, those which form direct nociceptive relays, rather than considering the swathe of physiological processes which are also entwined and govern our overall pain percept. Whilst there are numerous studies exploring the integration of acute pain and autonomic activity (e.g. Burton et al., 2009), which is vital for defensive behaviour expression, few studies have explored the integration of autonomic function with chronic pain. Throughout their publication, Venezia et al. (2024) not only acknowledge, but probe into how the autonomic nervous system (ANS) is involved in the perception and modulation of pain, both in chronic pain sufferers and in pain-free individuals.

Their findings reveal a fundamental difference in how activation of the ANS, through baroreceptor stimulation, alters perceived pain between chronic pain sufferers and those without pain, with their stimulation technique increasing and decreasing pain, respectively. Moreover, by conducting a conditioned pain modulation (CPM) task, a method for activating the brain's descending analgesic system also known as the 'pain inhibits pain' phenomenon, the authors demonstrate how resting tone of the ANS is directly linked to the magnitude of CPM expressed whilst conducting baroreceptor stimulation.

Venezia and colleagues present an elegant design, including a large sample size, to investigate behavioural effects (22 chronic pain; 29 no pain). Across a single study session, where autonomic activity (heart rate and blood pressure) was measured continuously, participants underwent a pressure pain calibration protocol to first adjust to inter-individual differences in pain thresholds, before undertaking an autonomic baseline, autonomic stress and pain task where, whilst pressure pain was delivered to the right fingernail, either sham or active baroreceptor stimulation occurred through suction of the left and right carotid bifurcation. During a final task period, pain was first delivered on a longer time scale to the left fingernail, before a short-lasting pressure pain was delivered simultaneously to the right fingernail, triggering the CPM response.

With particular interest being placed in recent years in uncovering biomarkers to chronic pain persistence and magnitude, the findings presented by Venezia and colleagues provide a unique lens into the underlying pathophysiology of chronic low back pain. Specifically, they identify relationships between resting blood pressure and the intensity of reported pain both at the time of undertaking the experimental session and over the preceding year prior. These relationships, whilst novel, are supported through the prominent theory that chronic pain states are driven through dysfunction in brainstem circuitry (Mills et al., 2018), with these same circuits playing a core role in ANS functions – particularly the midbrain periaqueductal grey matter. It is possible that the dysfunctional brainstem tone underlying chronic pain extends beyond sensory symptoms, and alters other core functions such as blood pressure and heart rate variability. Following this interpretation, the divergent findings the authors present as to how baroreceptor activation, when compared to sham stimulation, decreased and increased reported pain between no pain and chronic pain sufferers exposes another potential mechanistic shift in the function of this circuitry. That is, if in chronic pain sufferers this core circuitry which governs physiological and pain perceptual processes is dysfunctional, additional stimulation to this circuitry facilitates perceived pain further. Conversely, when these circuits are intact (e.g. in pain-free individuals), stimulation of the ANS causes activation within naturally inhibitory brainstem circuits, reducing perceived pain.

It is of note, however, that despite these divergent effects on pain *perception*, the authors found no similar group-level effects on pain *modulation*, and CPM responses were similar between groups when baroceptor activation occurred. However, here the authors rather elegantly shift their focus and identify another potential biomarker underlying the CPM response more generally: resting ANS tone. Taking the previous interpretation that the ANS and descending analgesic systems share common nodes, this finding is of particular relevance, as to date most literature suggests a core node of the brainstem – the subnucleus reticularis dorsalis – as critical in the human expression of CPM (Youssef et al., 2016). Not only does this nucleus reside within the medulla where several ANS nuclei are found, but it also reciprocally connects with these same nuclei (Martins & Tavares 2017), providing a route by which ANS activity and CPM responses may be linked.

When conducting pain studies, it is often easier to consider perceived pain in a vacuum, with brain function and behaviours measured solely as a reflection of that perception. However, decades of neuroscientific research have established the pain percept as a complex interplay between different biological systems. The findings of Venezia and colleagues represent a core step in our understanding of how these systems may link, and by extension how their measurement may help us better understand, interpret and manage the sensory symptoms of chronic pain.

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

## Additional information

### Competing interests

The authors declare no conflict of interest.

### Author contributions

Luke Henderson: Conception or design of the work; Drafting the work or revising it critically for important intellectual content; Final approval of the version to be published; Agreement to be accountable for all aspects of the work.

### Funding

Federal Government | DHAC | National Health and Medical Research Council (NHMRC): Luke A Henderson, 1130280.

### Keywords

baroreceptor reflex, chronic pain, pain perception, periaqueductal grey matter

## Supporting information

Additional supporting information can be found online in the Supporting Information section at the end of the HTML view of the article. Supporting information files available:

**Peer Review History**

