## [Peer Review History · The Journal of Physiology]

When pain is not the same: Baroreceptor activation delineates chronic pain and pain free participants perception and modulation of noxious pressure

Luke A Henderson

DOI: 10.1113/JP287685

Corresponding author(s): Luke Henderson (luke.henderson@sydney.edu.au)

Review Timeline:

Submission Date:	19-Sep-2024
Editorial Decision:	03-Oct-2024
Revision Received:	11-Apr-2025
Editorial Decision:	16-Apr-2025
Revision Received:	23-Apr-2025
Accepted:	28-Apr-2025

Senior Editor: Richard Carson

Reviewing Editor: Vaughan Macefield

Transaction Report:

Dear Dr Henderson,

Re: JP-P-2024-287685 "When pain is not the same: Baroreceptor activation delineates chronic pain and pain free participants perception and modulation of noxious pressure" by Luke A Henderson

Thank you for submitting your manuscript to The Journal of Physiology. It has been assessed by a Reviewing Editor and by 1 expert referee and we are pleased to tell you that it is acceptable for publication following satisfactory revision.

Please address all the points raised and incorporate all requested revisions or explain in your Response to Referees why a change has not been made. We hope you will find the comments helpful and that you will be able to return your revised manuscript within 4weeks. If you require longer than this, please contact journal staff: jp@physoc.org.

REVISION CHECKLIST:

- 'Potential Cover Art' for consideration as the issue's cover image
- Appropriate Supporting Information (Video, audio or data set: see https://jp.msubmit.net/cgi-bin/main.plex?form_type=display_requirements#supporting_information)

form_type=display_requirements#supp).

We look forward to receiving your revised submission.

Yours sincerely,

Richard Carson
Senior Editor
The Journal of Physiology

REQUIRED ITEMS FOR REVISION

Title page gives all the details of the FOCUS paper

This is confusing as in fact the Perspective author details are all missing...

Please correct this in a revised version

EDITOR COMMENTS

Reviewing Editor:

Thank you for submitting your Perspectives article. As you will see, the authors of the original article are satisfied that your article provides a realistic interpretation of their findings and have no recommendations for improvement. However, before it can be accepted could you please add the citation details of the Venezia et al. article to your reference list. Also, please refer to the article by Youssef and colleagues as Youssef et al. within the text.

REFEREE COMMENTS

Referee #1:

We would like to thank you for taking the time to read our article and for the insightful comments that support the core aspects of our research.

No misinterpretations have been found in the Perspective.

On the contrary, we support the your statement regarding the shared nodes between different systems, such as the autonomic nervous system and descending inhibitory pathways. Our previous MRI study on pain-free individuals demonstrated that the association between Heart Rate Variability and Conditioned Pain Modulation is mediated by the periaqueductal gray (Makovac et al., 2021). This region, along with the ventrolateral medulla, plays a significant role in the etiology and maintenance of chronic pain (De Felice et al., 2011; Hemington & Coulombe, 2015).

END OF COMMENTS

Dear Editor,

The attached file is in response to an invitation to write a commentary on the published article outlined below. I appreciate it is late, but I thought it was already submitted. If it is too late, I can withdraw the article.

Regards

Luke Henderson

Title: Investigating the Effects of Artificial Baroreflex Stimulation on Pain Perception: A Comparative Study in No-pain and Chronic Low Back Pain Individuals

Authors: Alessandra Venezia¹, Harriet-Fawsitt Jones¹, David Hohenschurz-Schmidt², Matteo Mancini³, Matthew Howard¹, Elena Makovac^{1,4*}

Affiliations:

¹ *Department of Neuroimaging, Institute of Psychology, Psychiatry & Neuroscience, King's College London, London, UK*

² *Department of Surgery & Cancer, Imperial College London, London, UK*

³ *Department of Cardiovascular, Endocrine-Metabolic Diseases and Aging, Italian National Institute of Health, Rome, Italy*

⁴ *Department of Life Sciences, Division of Psychology, Brunel University London, London, UK*

***Corresponding Author:** Dr Elena Makovac, Brunel University London, Division of Psychology, Department of Life Sciences, Centre for Cognitive and Clinical Neuroscience, Uxbridge, UB8 3PH, UK.

e-mail: elena.makovac@brunel.ac.uk

Dear Dr Henderson,

Re: JP-P-2025-287685R1 "When pain is not the same: Baroreceptor activation delineates chronic pain and pain free participants perception and modulation of noxious pressure" by Luke A Henderson

Thank you for submitting your manuscript to The Journal of Physiology. It has been assessed by a Reviewing Editor and we are pleased to tell you that it is acceptable for publication following satisfactory revision.

The review comments are copied at the end of this email.

Please address all the points raised and incorporate all requested revisions or explain in your Response to Referees why a change has not been made. We hope you will find the comments helpful and that you will be able to return your revised manuscript within 2 weeks. If you require longer than this, please contact journal staff: jp@physoc.org.

REVISION CHECKLIST:

We look forward to receiving your revised submission.

Yours sincerely,

Richard Carson
Senior Editor
The Journal of Physiology

REQUIRED ITEMS

- Please include a full, separate title page as part of your main article (Word) file, which should contain the following: title, authors, affiliations, corresponding author name and contact details, keywords, and running title.

EDITOR COMMENTS

Reviewing Editor:

Thank you for submitting your revised manuscript to The Journal of Physiology. I am satisfied with the minor amendments you have already made but there is just a minor grammatical fix you will need to do before I can recommend acceptance:

Please change "...participants perception..." to "...participants' perception..." in the manuscript file and in the journal metadata.

END OF COMMENTS

Dear Editor,

The attached file is a revised commentary on the published article outlined below.

Regards

Luke Henderson

Title: Investigating the Effects of Artificial Baroreflex Stimulation on Pain Perception: A Comparative Study in No-pain and Chronic Low Back Pain Individuals

Authors: Alessandra Venezia¹, Harriet-Fawsitt Jones¹, David Hohenschurz-Schmidt², Matteo Mancini³, Matthew Howard¹, Elena Makovac^{1,4*}

Affiliations:

¹ *Department of Neuroimaging, Institute of Psychology, Psychiatry & Neuroscience, King's College London, London, UK*

² *Department of Surgery & Cancer, Imperial College London, London, UK*

³ *Department of Cardiovascular, Endocrine-Metabolic Diseases and Aging, Italian National Institute of Health, Rome, Italy*

⁴ *Department of Life Sciences, Division of Psychology, Brunel University London, London, UK*

***Corresponding Author:** Dr Elena Makovac, Brunel University London, Division of Psychology, Department of Life Sciences, Centre for Cognitive and Clinical Neuroscience, Uxbridge, UB8 3PH, UK.

e-mail: elena.makovac@brunel.ac.uk

Dear Professor Henderson,

Re: JP-P-2025-287685R2 "When pain is not the same: Baroreceptor activation delineates chronic pain and pain free participants perception and modulation of noxious pressure" by Luke A Henderson

We are pleased to tell you that your paper has been accepted for publication in The Journal of Physiology.

Yours sincerely,

Richard Carson
Senior Editor
The Journal of Physiology

If you would like to receive our 'Research Roundup', a monthly newsletter highlighting the cutting-edge research published in The Physiological Society's family of journals (The Journal of Physiology, Experimental Physiology, Physiological Reports, The Journal of Nutritional Physiology, and The Journal of Precision Medicine: Health and Disease), please click this link, fill in your name and email address and select 'Research Roundup':

<https://www.physoc.org/journals-and-media/membernews>

- You can help your research get the attention it deserves! Check out Wiley's free Promotion Guide for best-practice recommendations for promoting your work at: www.wileyauthors.com/eo/guide. You can learn more about Wiley Editing Services which offers professional video, design, and writing services to create shareable video abstracts, infographics, conference posters, lay summaries, and research news stories for your research at: www.wileyauthors.com/eo/promotion.

The Corresponding Author will receive an email from Wiley with details on how to register or log-in to Wiley Authors Services where you will be able to place an order

EDITOR COMMENTS

Senior Editor:

Comments to the Author:

The author seems not to have made the single change requested, i.e., to add an apostrophe following "participants" in the title.

We will ask the copy editors to correct this at proof stage.